# Immobilized Sulfuric Acid on Silica Gel as Highly Efficient and Heterogeneous Catalyst for the One-Pot Synthesis of Novel α-Acyloxycarboxamides in Aqueous Media

**DOI:** 10.3390/ijms23179529

**Published:** 2022-08-23

**Authors:** Sodeeq Aderotimi Salami, Meloddy Manyeruke, Xavier Siwe-Noundou, Rui Werner Maçedo Krause

**Affiliations:** 1Department of Chemistry, Rhodes University, Grahamstown 6140, South Africa; 2Department of Pharmaceutical Sciences, School of Pharmacy, Sefako Makgatho Health Sciences University, Pretoria 0204, South Africa

**Keywords:** passerini reaction, aqueous media, α-acyloxycarboxamides, immobilized sulfuric acid on silica gel

## Abstract

The application of immobilized sulfuric acid on silica gel (H_2_SO_4_-SiO_2_) as an efficient and easily reusable solid catalyst was explored in the synthesis of novel α-acyloxycarboxamide derivatives via a Passerini reaction of benzoic acid, aldehyde/ketone, and isocyanides. The Passerini adducts were obtained in high to excellent yields within 10 min in aqueous media under catalytic conditions. The key advantages of the process include a short reaction time, high yields, the catalyst’s low cost, and the catalyst’s reusability.

## 1. Introduction

Multicomponent reactions (MCRs) involve more than two starting components to produce a new product that incorporates structural features of each reagent [1]. MCRs have the advantage of simplicity and synthetic efficiency over conventional chemical reactions, in addition to developing structural complexity in a single step. Selectivity, synthetic convergency, and atom economy are all advantages of the most useful MCRs [2,3]. MCRs are the foundations of both combinatorial chemistry and diversity-oriented synthesis and have thus played a key role in the development of current synthetic methodologies for pharmaceutical and drug discovery research [4]. Combinatorial chemistry approaches can be utilized to introduce or broaden structural variations in a target lead compound [5]. Therefore, diversity-oriented synthesis is valuable for probing large areas of chemical structure space in the search for new bioactive small molecules that may be undetected by traditional natural product screening tests. The two approaches are complementary, and they both contribute to the ability of MCRs to generate structural complexity [6].

Isocyanide-based MCRs have become a common organic chemical process in the pharmaceutical sector for the production of compound libraries of low-molecular-weight drug-like molecules. Isocyanide-based MCRs have been frequently used in the Ugi and Passerini reactions [7]. The range of bond-forming mechanisms, their functional group tolerance, and the high levels of chemo-, regio-, and stereoselectivity typically observed in isocyanides all contribute to their considerable potential for multicomponent reaction development [8]. The Passerini three-component reaction (P-3CR), first reported in 1921, is a multicomponent reaction involving an isocyanide, an aldehyde, and a carboxylic acid to produce, in one step, an esterified hydroxycarboxamide, also known as a depsipeptide, an essential intermediate in organic synthesis [9]. Medicinal chemists, particularly those working in the discipline of combinatorial chemistry, frequently use multicomponent reactions such as the Passerini because huge libraries of compounds may be rapidly synthesized from a commercial pool of starting materials [10]. This class of molecules is also present in many natural products and pharmacologically active depsipeptides, including enterochelin and pumilacidins [7].

The Passerini reaction has also been utilized to synthesize a variety of valuable structural motifs, including cyclic lipopeptides, which are antibacterial natural products with a broad spectrum of biological activities [11]. In just two steps, the Passerini reaction also yields the fungicidal compound mandipropamid, which is formed by the reaction of an in situ-synthesized isocyanide, an aldehyde, and a carboxylic acid (Figure 1). Micora (mandipropamid) is obtained by alkylation with propargyl bromide in the second step [11,12].

Water, on the other hand, is a safe, nontoxic, low-cost, and environmentally friendly solvent for organic transformations. Since Breslow’s seminal work in 1980, a wide range of organic reactions have been demonstrated to take place in aqueous media, sometimes with notable improvements, such as faster reaction rates and higher selectivities, when compared to results obtained using traditional organic-solvent-based systems [13,14]. Water is the medium through which all life operates and has distinct physical and chemical properties. It is surprising that this medium, which has been mostly disregarded throughout the last 150–200 years of modern organic chemistry, is yielding novel and unexpected experimental results [1]. Using water as a solvent for organic transformations has several environmental advantages. As a result, water as a reaction medium has received a lot of attention in synthetic organic chemistry because it is a cheap, nontoxic, and nonflammable solvent. The use of water as a medium for organic reactions has properties that may be beneficial in a variety of reactions [15].

Furthermore, as an example of a one-pot multicomponent reaction, the formation of α-acyloxycarboxamides from a carboxylic acid, an aldehyde, and substituted isocyanide was investigated in aqueous media. Usually, Passerini reactions are performed in an organic solvent such as CH_2_Cl_2_ or MeOH, and they can take 24 h or more for completion [16]. In addition to the negative environmental impact of using organic solvents and the inconvenience of the longer reaction time, this also makes them impractical for use in a typical three-hour undergraduate laboratory session. In an attempt to find suitable conditions we could use to introduce MCR methodologies to our undergraduate classes, we substituted the organic solvent with water. Surprisingly, the Passerini reaction was completed in 15 min, and excellent yields were observed. This method is significantly greener than the classical Passerini reaction because water is an ideal nontoxic, nonflammable, environmentally friendly, cheap, and readily available solvent. In aqueous media, several organic processes have shown significant increases in the rates of reaction, even though their origins are still a mystery [17,18]. Water might not be the best solvent for many of these reactions because they are biphasic. However, heterogeneous aqueous reaction rates are frequently related to mixing speed and method, and these rates are frequently inversely proportional to reaction temperature [16]. The effects have been interpreted as a result of the aqueous media’s cohesive energy density, the hydrophobicity of the reagents, increased hydrogen bonding in the transition state, or a decrease in the transition state’s volume [19].

Over the past few decades, simple and practical examples of Passerini reactions have been used in a variety of transformations to uncover possible drug-like compounds. These reactions have occasionally been carried out in noncatalytic conditions using excessive amounts of reactants with low efficiency over lengthy periods of time, sometimes up to several days. In addition to diverse methods, the Passerini reaction has been broadened using a variety of catalysts, including Cu(II) [20], Al(III) [21], aluminum-organophosphate [22], SiCl_4_ [23], In(OTf)_3_ [24], Ti(i-PrO) [25], and tetramethylguanidine-functionalized silica nanoparticles (TMG-SiO_2_ NPs) [25]. These catalysts have a high degree of efficiency, and some of them are simple to separate, but they also have some drawbacks, including a high cost, toxicity, non-reusability, lengthy reaction times, environmental pollution, and difficult reaction conditions. As a result, it is crucial to develop new catalytic methodologies with minimal fouling effects, catalyst thermal stability, easy operation, and catalyst recovery processes for the Passerini reaction. 

As part of our ongoing research on the synthesis of novel organic compounds utilizing immobilized sulfuric acid on silica gel (H_2_SO_4_-SiO_2_) [26], we report the preparation of a new class of α-aryloxy amide derivatives by a novel three-component Passerini reaction of benzoic acid (**1a**), aldehydes (**2a**), and isocyanides (**3a**) in the presence of immobilized sulfuric acid on silica gel in aqueous media (Figure 1). A clean, effective, and environmentally safe chemical reaction to generate novel α-acyloxycarboxamide derivatives was carried out in the presence of immobilized sulfuric acid on silica gel. This heterogeneous catalyst was used in a Passerini reaction for the first time in aqueous media under mild conditions.

## 2. Results and Discussion

Multicomponent reactions have been characterized as efficient methods for synthesizing polyfunctional compounds with high diversity and complexity in a few steps. The reaction of benzoic acid (**1a**), benzaldehyde (**2a**), and 2-nitrophenylisocyanide (**3a**) was chosen as the model reaction, and they were subjected to a typical Passerini reaction in DCM at room temperature (18–35 °C) for 24 h (Table 1, entry 5). The desired product (**A**) was obtained in 85% yield.

To improve the efficiency of reaction, various reaction solvents were screened, such as diethyl ether, toluene, acetonitrile, ethyl acetate, DCM, ethanol, and methanol. Although the reaction occurs with all the solvents used, dichloromethane and methanol were the best among the tested solvents, producing 85 and 78% yields, respectively (Table 1).

When the reaction was carried out in diethyl ether, toluene, acetonitrile, ethyl acetate, and ethanol, low to moderate amounts of the product were observed, producing yields of 43, 65, 58, 32, and 74%, respectively which may have been due to solubility issues. (Table 1, entries 1–7). Therefore, DCM was found to be the best organic solvent for the current reaction, as it provided the desired product in 85% yield within 24 h (Table 1, entry 5). At this point, we thought of carrying out these reactions under green conditions. To our delight, when the solvent was replaced with water at room temperature, the Passerini reaction was completed in 15 min (Figure 2), and the product was obtained in 90% yield (Table 1, entry 8). It was also discovered that extending the reaction time had little or no influence on the yield, as 15 min was found to be sufficient for completing the reaction with the highest possible yield.

Isocyanide has been found to undergo slow spontaneous hydrolysis in the presence of water [27]. Following that observation, we then focused on the optimization of the water content. The reaction was explored by increasing the volume of water from 1 to 5 mL to acquire a better understanding of the effect of water on the model Passerini reaction. The best result, of 90% yield, was observed when 2 mL of water by volume was introduced to the reaction (Table 2, entry 2). The effectiveness of the reaction declined dramatically when the amount of water was reduced (Table 2, entry 1), while increasing the amount of water also resulted in a lower product yield (Table 2, entries 3–5).

To investigate the effect of a catalyst on the model Passerini reaction and to establish the best catalytic conditions, we started the condensation of benzoic acid (1 mmol), benzaldehyde (1 mmol), and 2-nitrophenylisocyanide (1 mmol) in the presence of H_2_SO_4_ immobilized on SiO_2_ (0.01 g) as a catalyst at room temperature (18–35 °C) in aqueous media. The reaction furnished the corresponding Passerini adducts in 10 min, and a good yield (93%) of the product was obtained.

To enhance the reaction yields of the products, catalytic studies were carried out by increasing the catalyst amount from 0.01 g to 0.05 g at room temperature (18–35 °C) in aqueous media while keeping the other experimental parameters constant. The conversion rate gradually increased from 93% to 98% within 10 min when the catalyst amount was increased to 0.02 g (Figure 3). This is because more acidic sites became available as the catalyst concentration increased. The yield was unaffected by further increasing the catalyst loading (0.03–0.04 g). Further loading with 0.05 g of catalyst had a detrimental effect on the yield. As the catalyst amount was increased, the percentage conversion increased to a high point before gradually declining. An excessive amount of catalyst may obstruct conversion due to catalyst poisoning caused by an excessive amount of reactants adhering to the catalyst surface. The amount of catalyst determines the availability of acidic sites, which when present in excess, may produce undesirable byproducts or promote the occurrence of the reversible reaction [28]. When the reaction was carried out in the presence of silica gel (without catalyst) the Passerini adducts were obtained in 64% yield within 15 min, which implies that the reaction rate depends greatly on the catalyst used. The optimal quantity of catalyst for the successful Passerini reaction in aqueous media was determined to be 0.02 g.

To demonstrate the catalytic significance of H_2_SO_4_-SiO_2_ for this reaction, we applied this catalyst to synthesize α-acyloxycarboxamide derivatives under aqueous conditions utilizing a variety of aromatic isocyanides with a wide range of ortho-, meta-, and para-substitutions. Generally, the synthetic procedure involved stirring the mixture of benzoic acid (**1a**, 1 mmol), benzaldehyde (**2a**, 1 mmol), isocyanide (**3a**, 1 mmol), and H_2_SO_4_ immobilized on SiO_2_ (0.02 g) in water (2 mL) for 10 min at room temperature (18–35 °C). The corresponding results are given in Figure 2. We discovered that the reaction worked exceptionally well with either electron-releasing or electron-withdrawing substituents on the aryl ring of the isocyanide. Various functional groups, such as fluoro, chloro, methyl, and methoxy groups, were found to be well-tolerated.

The obtained α-acyloxycarboxamide derivatives (A-R) (Figure 2) were confirmed based on their NMR spectral data (Appendix A). The ^1^H NMR spectra of this series showed NH signals ranging from δH 8.00 to 11.00 ppm. The aromatic protons of carboxylic acid and aldehyde moieties also appeared in the expected range (δH 7.80–6.40 ppm). ^13^C NMR spectra exhibited signals ranging from δC 161.0 to 173.0 ppm, corresponding to the two amido-ester C=O groups.

The Passerini reaction under the above optimized condition did not afford the expected bis α-acyloxycarboxamide derivatives in the case of phthalic acid (**1b**) with **2a** and **3a**. This may be attributed to the planarity of the phenyl ring preventing the second Passerini reaction due to steric hindrance. According to a literature review, succinic acid could be used to synthesize bis α-acyloxycarboxamides [28]. The free rotation of the C-C bond between the two methylene groups was suggested as a strategy to increase the flexibility and avoid steric restriction around the -COOH. However, the synthesis of bis α-acyloxycarboxamides utilizing succinic acid (**1c**) under the above experimental condition was unsuccessful; all trials led to the hydrolysis of the corresponding isocyanide (Figure 4).

The versatility of the standardized Passerini reaction conditions for the synthesis of novel α-acyloxycarboxamides was also examined for the carbonyl component of the Passerini reaction while employing 5-bromo isatin (**2a**) (5-bromo Indole 2,3-dione) as the carbonyl component and 4-bromo benzoic acid (**1c**) (Figure 5). It was observed that the developed reaction conditions were robust for accepting indole 2,3-dione. Thus, the reaction of isatin (**2b**), benzoic acid (**1a**), and 3-methoxyphenylisocyanide (**3b**) under optimized reaction conditions led to the formation of (3-methoxyphenylcarbamoyl)(2-oxoindolin-3-yl)methyl benzoate in 90% yield (Figure 6).


**Utility of 4-chlorophenol as an acid surrogate**


The synthetic potential of this protocol was examined by utilizing electron-deficient phenols as the acid component in the reaction. Thus, the Passerini reaction involving isatins (**2c**) with 4-chlorophenol (**1b**) and 2-nitrophenylisocyanide (**3a**) under optimized reaction conditions resulted in the synthesis of an O-arylated oxindole derivative in good to excellent yields (Figure 5). In this reaction, the irreversible Smiles rearrangement of the intermediate phenoxyimidate adducts resulted in the formation of the O-arylated product, and this is an example of the application of the Smiles rearrangement in a Passerini reaction.

We also studied the recyclability and reusability effect of a H_2_SO_4_·SiO_2_ catalyst in these reactions. After each reaction, the catalyst was filtered, cleaned with ethyl acetate, dried at 120 °C for three hours, and then utilized in another cycle of reactions. The catalyst reusability demonstrated that it was extremely effective in the synthesis of Passerini products. After five cycles, the percentage conversion efficiency under the optimum conditions dropped from 98% to 87%. The higher catalytic efficiency was due to the higher acidity produced by the H-bond and hydrophobic core of the mesoporous catalyst (Table 3).

Figure 7 illustrates the proposed mechanism for the formation of α-acyloxycarboxamide derivatives in the presence of SiO_2_-H_2_SO_4_. The catalyst first activates the carbonyl group of the aldehydes (**4**). Then, the isocyanide attacks the activated carbonyl (**5**) group nucleophilically, forming an intermediate known as nitrilium (**6**). The intermediate (**6**) is then attacked by the carboxylate of (**7**), which is followed by an acyl transfer by a Mumm rearrangement to generate derivatives of oxindole-based acyloxycarboxamides (**9**).


**Characterization of the catalyst**



**FT-IR spectral analysis**


The FT-IR spectrum of the catalyst is shown in Figure 2. The catalyst was solid, and the solid-state IR spectrum was obtained by the attenuated total reflection (ATR) method. For each experiment, 16 scans were performed in the frequency range from 650 to 4000 cm^−1^. The wide bands detected in the spectra (**A**) at 3511 and (**C**) 3486 cm^−1^ were attributed to superimposed stretching modes of Si–OH groups and hydroxyl groups (O–H). The peaks from Si–O–Si groups, which occur in the range of 1250–1000 cm^−1^ for antisymmetric stretching, are a characteristic indicator of the silica structure. For silica (**B**, SiO_2_) without a catalyst, the major peaks were broad antisymmetric Si–O–Si stretching from 1200 to 1000 cm^−1^ and symmetric Si–O–Si stretching near 800 cm^−1^. 


**Powder X-ray diffraction (XRD)**


As shown in Figure 3, the XRD pattern of (A) the immobilized sulfuric acid on silica gel displayed a noticeable signal at a 2*θ* angle of 22.91° (a broad diffraction band), which is reminiscent of a typical amorphous silica diffraction pattern. The same broad diffraction band was noticed for the recovered catalyst (C) after five cycles (Figure 3C). A certain conclusion can be made about the crystalline structure of the support, i.e., silica was not distorted during the preparation of the catalysts.

## 3. Experimental Section

### 3.1. Materials and Methods

A PerkinElmer Spectrum 100 FT-IR Spectrometer was used for the FT-IR analysis. The IR spectra were obtained by the attenuated total reflection (ATR) method. For each experiment, 16 scans were performed in the frequency range from 650 to 4000 cm^−1^. The melting points of all the compounds were determined using a Koffler hot-stage apparatus and are uncorrected. The NMR spectra were recorded on a Bruker Advance III 400 spectrometer using CDCl_3_ or DMSO-d_6_ as a solvent with tetramethyl silane used as an internal standard. LC-MS/MS data were recorded on a Bruker Compact quadrupole time of flight (QToF) mass spectrometer. Raw mass spectrometry data were processed using MZmine free online software (version 2.38). Powder X-ray diffraction measurements were performed using a D8 Advance diffractometer made by a Bruker AXS company in Germany. The solvents and chemicals used were of analytical grade, purchased from Sigma Aldrich, and used without further purification. The purity determination of the starting materials and reaction monitoring were performed by thin-layer chromatography (TLC) on Merck silica gel G F254 plates. 

### 3.2. Preparation of Sulfuric Acid Adsorbed on Silica Gel (SiO_2_-H_2_SO_4_) 

This was previously reported [29]. To a suspension of silica gel (29.5 g, 230–400 mesh size) in EtOAc (60 mL), H_2_SO_4_ (1.5 g, 15.5 mmol, 0.8 mL of a 98% aq. solution of H_2_SO_4_) was added, and the mixture was stirred magnetically for 30 min at room temperature. EtOAc was removed under reduced pressure (rotary evaporator), and the residue was heated at 100 °C for 72 h under vacuum to afford SiO_2_-H_2_SO_4_ as a free-flowing powder.


**General Experimental Procedures**


A mixture of benzoic acid (0.122 g, 1 mmol), benzaldehyde (0.10 mL, 1 mmol), and isocyanide (0.15 g 1 mmol) was vigorously stirred in 2 mL of water at room temperature for 10 min in the presence of 0.02 g of immobilized sulfuric acid on silica gel (SiO_2_-H_2_SO_4_). Upon completion, the organic layer was separated and collected by a separating funnel. The combined organic phases were concentrated under reduced pressure, and the residue was purified by column chromatography using DCM/hexane (3:1) as an eluent to afford the desired products. Typical yields ranged from 60 to 98%. All other products (A-X) were obtained by a similar approach.
**A.** **(2-nitrophenylcarbamoyl)(phenyl)methyl benzoate**

Prepared from 1-isocyano-2-nitrobenzene (0.15 g, 1 mMol), benzaldehyde (0.11 mL, 1 mMol), and benzoic acid (0.122 g, 1 mMol,) according to the general procedure. Purification: column chromatography on silica gel (3:1 DCM/hexane). Isolated as a white powder (98% yield); mp 112–114 °C; 1H NMR (400 MHz, CDCl_3_) δ 10.26 (s, 1H, NH), 8.71 (d, J = 8.4 Hz, 1H, Ar-H), 8.51 (s, 1H, Ar-H), 8.15 (d, J = 9.8 Hz, 1H, Ar-H), 8.04 (d, J = 8.4 Hz, 3H, Ar-H), 7.64–7.48 (m, 2H, Ar-H), 7.39 (t, J = 7.7 Hz, 3H, Ar-H), 7.26 (t, J = 7.7 Hz, 1H, Ar-H), 7.14 (t, J = 7.8 Hz, 1H, Ar-H), 6.72 (d, J = 8.4 Hz, 1H, Ar-H), 6.60 (t, J = 7.2 Hz, 1H, (O-(CH)CO). 13C NMR (101 MHz, CDCl_3_) δ 172.40, (NH-CO), 159.83, (O-C=O), 144.69, 136.24, 135.71, 133.88, 133.63, 130.23, 129.33, 128.54, 126.21, 125.91, 124.07, 122.81, 118.77, 116.86. HR-MS (ESI): [M+H^+^] calculated for C_21_H_17_N_2_O_5_: 377.1211, found: 377.2603.
**B.** **(2-bromophenylcarbamoyl)phenyl)methyl benzoate**

Prepared from 1-bromo-2-isocyanobenzene (0.18 g, 1 mMol), benzaldehyde (0.11 mL, 1 mMol), and benzoic acid (0.122 g, 1 mMol,) according to the general procedure. Purification: column chromatography on silica gel (3:1 DCM/hexane). Isolated as a white powder (76% yield); mp 100–102 °C; IR (NaCl) v(cm^−1^) 3253 (NH), 1712 (CO) ester, and 1628 (CO) amide. 1H NMR (400 MHz, CDCl_3_) δ 11.18 (s, 1H), 8.31 (d, J = 8.2 Hz, 1H, NH), 8.13 (d, J = 7.3 Hz, 1H, Ar-H), 8.03 (d, J = 7.2 Hz, 2H, Ar-H), 7.52 (dd, J = 15.8, 8.3 Hz, 1H, Ar-H), 7.44–7.29 (m, 3H, Ar-H), 7.20 (t, J = 7.7 Hz, 1H, Ar-H), 6.89 (t, J = 8.3 Hz, 1H, Ar-H), 6.45 (s, 1H, O-(CH)CO). 13C NMR (101 MHz, CDCl_3_) δ 166.75, (NH-CO), 164.62, (O-C=O), 135.18, 134.84, 134.00, 133.90, 132.28, 130.47, 130.27, 130.04, 129.39, 129.35, 129.07, 128.86, 128.77, 128.62, 128.55, 128.19, 128.16, 127.44, 125.79, 121.67, 113.53, 76.18 (O-(CH)CO). HR-MS (ESI): [M+H^+^] calculated for C_21_H_16_BrNO_3_: 410,0325, found: 410.0331.
**C.** **(2-chlorophenylcarbamoyl)(phenyl)methyl benzoate**

Prepared from 1-chloro-2-isocyanobenzene (0.14 g, 1 mMol), benzaldehyde (0.11 mL, 1 mMol), and benzoic acid (0.122 g, 1 mMol) according to the general procedure. Purification: column chromatography on silica gel (3:1 DCM/hexane). Isolated as a white powder (96% yield); mp 143–145 °C; IR (NaCl) v(cm^−1^) 3300 (NH), 1715 (CO) ester, and 1675 (CO) amide. 1H NMR (400 MHz, CDCl_3_) δ 8.66 (s, 1H, NH), 8.35 (d, J = 8.2 Hz, 1H, Ar-H), 8.12 (d, J = 7.4 Hz, 2H, Ar-H), 7.56 (t, J = 8.0 Hz, 3H, Ar-H), 7.43 (t, J = 7.7 Hz, 2H, Ar-H), 7.37 (s, 1H, Ar-H), 7.33 (dd, J = 9.8, 7.3 Hz, 3H, Ar-H), 7.28 (d, J = 8.0 Hz, 1H, Ar-H), 7.20 (s, 1H, Ar-H), 7.19–7.15 (m, 1H, Ar-H), 6.97 (dd, J = 11.2, 4.3 Hz, 1H, Ar-H), 6.44 (s, 1H, (O-(CH)CO). 13C NMR (101 MHz, CDCl_3_) δ 166.45, (NH-CO), 164.35, (O-C=O), 135.27, 133.92, 133.84, 129.92, 129.32, 129.01, 128.93, 128.75, 127.94, 127.37, 125.16, (Ar-Cl), 122.88, 121.33, 76.18 (O-(CH)CO). HR-MS (ESI): [M+H^+^] calculated for C_21_H_16_ClNO_3_: 366.0831, found: 366.1302.
**D.** **(3,4-difluorophenylcarbamoyl)(phenyl)methyl benzoate**

Prepared from 1,2-difluoro-4-isocyanobenzene (0.14 g, 1 mMol), benzaldehyde (0.11 mL, 1 mMol), and benzoic acid (0.122 g, 1 mMol) according to the general procedure. Purification: column chromatography on silica gel (3:1 DCM/hexane). Isolated as a white powder (66% yield); mp 150–151 °C; IR (NaCl) v(cm^−1^) 3273 (NH), 1732 (CO) ester, and 1671 (CO) amide. 1H NMR (400 MHz, CDCl_3_) δ 8.49 (s, 1H, (NH), 8.17 (d, J = 7.7 Hz, 1H, Ar-H), 8.11 (d, J = 8.3 Hz, 1H, Ar-H), 7.72–7.57 (m, 2H, Ar-H), 7.52 (dd, J = 12.6, 4.8 Hz, 1H, Ar-H), 7.46 (dd, J = 10.9, 5.5 Hz, 2H, Ar-H), 7.15–6.98 (m, 1H, Ar-H), 6.45 (s, 1H, (O-(CH)CO).). 13C NMR (101 MHz, CDCl_3_) δ 166.46, (NH-CO), 165.40, (O-C=O), 151.01 (NH-R-C-F), 148.91, 134.67, 134.00, 133.71, 130.19, 129.94, 129.47, 129.06, 128.89, 128.78, 128.50, 127.47, 117.31, 117.13, 110.07, 109.85, 76.13, (O-(CH)CO). HR-MS (ESI): [M+H^+^] calculated for C_21_H_15_F_2_NO_3_: 368,1015, found: 368.1022.
**E.** **(3,4-dichlorophenylcarbamoyl)phenyl)methyl benzoate**

Prepared from 1,2-dichloro-4-isocyanobenzene (0.17 g, 1 mMol), benzaldehyde (0.11 mL, 1 mMol), and benzoic acid (0.122 g, 1 mMol) according to the general procedure. Purification: column chromatography on silica gel (3:1 DCM/hexane). Isolated as a yellow solid (73% yield); mp 152–153 °C; IR (NaCl) v(cm^−1^) 3273 (NH), 1722 (CO) ester, and 1671 (CO) amide. 1H NMR (400 MHz, CDCl_3_) δ 8.15 (s, 1H, NH), 8.05 (t, J = 8.2 Hz, 3H, Ar-H), 7.65 (s, 1H, Ar-H), 7.54 (t, J = 7.1 Hz, 3H, Ar-H), 7.41 (dd, J = 15.7, 7.9 Hz, 4H, Ar-H), 7.34 (d, J = 6.4 Hz, 2H, Ar-H), 7.27–7.14 (m, 2H, Ar-H), 6.34 (s, 1H, O-(CH)CO). 13C NMR (101 MHz, CDCl_3_) δ 166.75, (NH-CO), 165.68, (O-C=O), 136.54, 134.51, 134.04, 133.75, 132.86, 130.46, 130.22, 129.98, 129.53, 129.33, 129.09, 128.84, 128.79, 128.51, 128.07, 127.50, 121.72, 119.24, 76.25, (O-(CH)CO). HR-MS (ESI): [M+H^+^] calculated for C_21_H_15_Cl_2_NO_3_: 427.0346, found: 427.1105.
**F.** **(3-cyanophenylcarbamoyl)phenyl)methyl benzoate**

Prepared from 3-isocyanobenzonitrile (0.13 g, 1 mMol), benzaldehyde (0.11 mL, 1 mMol), and benzoic acid (0.122 g, 1 mMol) according to the general procedure. Purification: column chromatography on silica gel (3:1 DCM/hexane). Isolated as a white powder (89% yield); mp 136–137 °C; IR (NaCl) v(cm^−1^) 3350 (NH), 1730 (CO) ester, and 1687 (CO) amide. 1H NMR (400 MHz, CDCl_3_) δ 11.30 (s, 1H, NH), 8.14–7.82 (m, 2H, Ar-H), 7.66–7.57 (m, 1H, Ar-H), 7.57–7.51 (m, 2H, Ar-H), 7.48 (t, J = 7.9 Hz, 1H, Ar-H), 7.40 (t, J = 7.7 Hz, 2H, Ar-H), 6.36 (s, 1H, O-(CH)CO). 13C NMR (101 MHz, CDCl_3_) δ 172.55, (NH-CO), 167.48, (O-C=O), 133.84, 132.96, 130.77, 130.24, 129.80, 129.40, 128.55, 127.62, 116.65, (C-CN), 113.96, 76.31, (O-(CH)CO). HR-MS (ESI): [M+H^+^] calculated for C_22_H_16_N_2_O_3_: 357.1172, found: 356.9086.
**G.** **(p-tolycarbamoyl)phenyl)methyl benzoate**

Prepared from 1-isocyano-4-methylbenzene (0.12 g, 1 mMol), benzaldehyde (0.11 mL, 1 mMol), and benzoic acid (0.122 g, 1 mMol) according to the general procedure. Purification: column chromatography on silica gel (3:1 DCM/hexane). Isolated as a white powder (94% yield); mp 148–149 °C; IR (NaCl) v(cm^−1^) 3285 (NH), 1730 (CO) ester, and 1642 (CO) amide. 1H NMR (400 MHz, DMSO) δ 10.44 (s, 1H, NH), 8.07 (s, 2H, Ar-H), 7.96 (s, 1H, Ar-H), 7.71 (s, 3H, Ar-H), 7.57 (s, 3H Ar-H), 7.48 (s, 6H, Ar-H), 7.10 (s, 2H, Ar-H), 6.26 (s, 1H, O-(CH)CO), 2.23 (s, 3H, CH3). 13C NMR (101 MHz, DMSO) δ 167.82 (NH-C=O), 166.82 (O-C=O), 136.42, 135.78, 134.23, 133.31, 133.22, 131.26, 129.93, 129.74, 129.66, 129.56, 129.37, 129.33, 129.17, 129.02, 127.86, 119.81, 76.30, (O-(CH)CO), 20.88 (CH3). HR-MS (ESI): [M+H^+^] calculated for C_22_H_19_NO_3_: 346.1318, found: 346.1315.
**H.** **(m-tolylcarbamoyl)(phenyl)methyl benzoate**

Prepared from 1-isocyano-3-methylbenzene (0.12 g, 1 mMol), benzaldehyde (0.11 mL, 1 mMol), and benzoic acid (0.122 g, 1 mMol) according to the general procedure. Purification: column chromatography on silica gel (3:1 DCM/hexane). Isolated as a yellow liquid (75% yield); IR (NaCl) v(cm^−1^) 3273 (NH), 1718 (CO) ester, and 1675 (CO) amide. 1H NMR (400 MHz, CDCl_3_) δ 8.08 (d, J = 7.4 Hz, 1H, (NH), 7.80 (s, 1H, Ar-H), 7.55 (dd, J = 14.0, 6.9 Hz, 2H, Ar-H), 7.43 (t, J = 7.7 Hz, 1H, Ar-H), 7.40–7.26 (m, 2H, Ar-H), 7.21 (d, J = 8.1 Hz, 1H, Ar-H), 7.11 (t, J = 7.8 Hz, 1H, Ar-H), 6.86 (d, J = 7.5 Hz, 1H), 6.37 (s, 1H, (O-(CH)CO), 2.23 (s, 2H, CH_3_), 2.09 (s, 1H, CH_3_). 13C NMR (101 MHz, CDCl_3_) δ 166.32, (NH-CO), 165.02, (O-C=O), 139.07, 136.84 (NH-R-CH3), 135.20, 133.83, 129.91, 129.25, 129.13, 128.96, 128.86, 128.75, 127.48, 125.71, 120.71, 117.13, 76.14, (O-(CH)CO), 21.43, (CH_3_). HR-MS (ESI): [M+H^+^] calculated for C_22_H_19_NO_3_: 346.1340, found: 346.1836.
**I.** **(3,5-dimethylphenylcarbamoyl)(phenyl)methyl benzoate**

Prepared from 3,5-dimethyl phenyl isocyanide (0.13 g, 1 mMol), benzaldehyde (0.11 mL, 1 mMol), and benzoic acid (0.122 g, 1 mMol) according to the general procedure. Purification: column chromatography on silica gel (3:1 DCM/hexane). Isolated as a white powder (94% yield); mp 148–149 °C; 1H NMR (400 MHz, CDCl_3_) δ 10.45 (s, 1H, NH), 8.03 (dd, J = 14.1, 7.5 Hz, 3H, Ar-H), 7.54 (t, J = 7.4 Hz, 2H, Ar-H), 7.40 (t, J = 7.7 Hz, 3H, Ar-H), 7.28 (d, J = 7.1 Hz, 1H, Ar-H), 6.95–6.84 (m, 1H, Ar-H), 6.82 (t, J = 7.4 Hz, 1H, Ar-H), 6.75 (d, J = 8.1 Hz, 1H, Ar-H), 6.32 (s, 1H, (O-(CH)CO), 4.44 (d, J = 5.9 Hz, 1H, CH3), 3.61 (s, 1H, CH3). 13C NMR (101 MHz, CDCl_3_) δ 168.04, NH(CO), 164.79, (O-C=O), 135.58, 133.77, 130.24, 129.83, 129.37, 129.10, 128.96, 128.75, 128.60, 128.51, 127.52, 125.52, 120.84, 110.27, 75.90, (O-(CH)CO), 55.11, 40.00. HR-MS (ESI): [M+H^+^] calculated for C_23_H_21_NO_3_: 361.1522, found: 361.1936.
**J.** **(mesitylcarbamoyl)(phenyl)methyl benzoate**

Prepared from 2-isocyano-1,3,5-trimethylbenzene (0.15 g, 1 mMol), benzaldehyde (0.11 mL, 1 mMol), and benzoic acid (0.122 g, 1 mMol) according to the general procedure. Purification: column chromatography on silica gel (3:1 DCM/hexane). Isolated as a colorless liquid (60% yield); IR (NaCl) v(cm^−1^) 3257 (NH), 1730 (CO) ester, and 1604 (CO) amide. 1H NMR (400 MHz, CDCl_3_) δ 8.10 (s, 1H), 8.05 (d, J = 7.5 Hz, 1H), 7.56 (t, J = 7.4 Hz, 1H), 7.42 (t, J = 7.8 Hz, 1H), 6.77 (s, 3H), 2.25 (s, 7H), 2.17 (s, 4H). 13C NMR (101 MHz, CDCl_3_) δ 166.96, 162.35, 138.81, 134.55, 131.32, 130.56, 129.61, 128.86, 128.46, 126.94, 21.17, 18.77. HR-MS (ESI): [M+H^+^] calculated for C_24_H_23_NO_3_: 374.1655, found: 374.2118.
**K.** **(4-methyl-2-nitrophenylcarbamoyl)phenyl)methylbenzoate**

Prepared from 4-methyl-2-nitro phenyl isocyanide (0.16 g, 1 mMol), benzaldehyde (0.11 mL, 1 mMol), and benzoic acid (0.122 g, 1 mMol) according to the general procedure. Purification: column chromatography on silica gel (3:1 DCM/hexane). Isolated as a yellow solid (92% yield); mp 126–129 °C; IR (NaCl) v(cm^−1^) 3343 (NH), 1694 (CO) ester, and 1641 (CO) amide. 1H NMR (400 MHz, CDCl_3_) δ 11.19 (s, 1H, NH), 8.60 (d, J = 8.6 Hz, 1H, Ar-H), 8.19 (d, J = 7.3 Hz, 2H, Ar-H), 7.95 (s, 1H, Ar-H), 7.56 (d, J = 7.7 Hz, 3H Ar-H), 7.35 (d, J = 7.6 Hz, 4H, Ar-H), 7.11 (d, J = 10.2 Hz, 2H, Ar-H), 6.66 (d, J = 8.5 Hz, 2H, Ar-H), 6.43 (s, 1H, O-(CH)CO), 2.42 (s, 3H, CH3). 13C NMR (101 MHz, CDCl_3_) δ 172.76, (NH-CO), 167.52, (O-C=O), 143.91, 142.77, 137.23, 136.37, 135.07, 134.39, 133.87, 131.89, 130.09, 129.56, 129.01, 128.77, 127.15, 126.62, 125.85, 125.71, 125.27, 122.02, 118.80, 76.36, (O-(CH)CO), 21.32. HR-MS (ESI): [M+H+] calculated for C_22_H_18_N_2_O_5_: 391.1383, found: 391.3062.
**L.** **(4-methoxyphenylcarbamoyl)phenyl)methyl benzoate**

Prepared from 1-isocyano-4-methoxy benzene (0.13 g, 1 mMol), benzaldehyde (0.11 mL, 1 mMol), and benzoic acid (0.122 g, 1 mMol) according to the general procedure. Purification: column chromatography on silica gel (3:1 DCM/hexane). Isolated as a yellow liquid (94% yield); IR (NaCl) v(cm^−1^) 3293 (NH), 1734 (CO) ester, and 1671 (CO) amide. 1H NMR (400 MHz, DMSO) δ 10.42 (s, 1H, NH), 8.08 (d, J = 7.2 Hz, 2H, Ar-NH), 7.70 (d, J = 7.3 Hz, 3H), 7.58 (t, J = 7.7 Hz, 2H), 7.52–7.46 (m, 4H), 7.43 (dd, J = 14.5, 7.4 Hz, 2H), 6.88 (d, J = 9.1 Hz, 2H), 6.22 (s, 1H, O-(CH)CO), 3.70 (s, 3H, OCH3). 13C NMR (101 MHz, DMSO) δ 166.57 (NHC=O), 165.69 (O-C=O), 155.95 (R-C-O-CH3), 135.81, 134.28, 132.01, 129.94, 129.74, 129.50, 129.37, 129.19, 129.03, 127.84, 121.25, 114.39, 76.27 (O-(CH)CO), 55.60 (OCH3). HR-MS (ESI): [M+H+] calculated for C_22_H_19_NO_4_: 362.1346, found: 362.1319.
**M.** **(2-methoxybenzylcarbamoyl)phenyl)methyl benzoate**

Prepared from 1-(isocyanomethyl)-2-methoxybenzene (0.15 g, 1 mMol), benzaldehyde (0.11 mL, 1 mMol), and benzoic acid (0.122 g, 1 mMol) according to the general procedure. Purification: column chromatography on silica gel (3:1 DCM/hexane). Isolated as a colorless liquid (94% yield); IR (NaCl) v(cm^−1^) 3273 (NH), 1722 (CO) ester, and 1663 (CO) amide. 1H NMR (400 MHz, CDCl_3_) δ 8.10–8.04 (m, 1H, NH), 8.02 (dd, J = 8.4, 1.2 Hz, 1H, Ar-H), 7.60–7.49 (m, 1H, Ar-H), 7.48–7.35 (m, 2H, Ar-H), 7.29 (d, J = 7.0 Hz, 1H, Ar-H), 7.23–7.12 (m, 1H, Ar-H), 6.83 (td, J = 7.4, 0.7 Hz, 1H, Ar-H), 6.76 (d, J = 8.2 Hz, 1H, Ar-H), 6.30 (s, 1H, O-(CH)CO)), 4.44 (d, J = 5.9 Hz, 1H, (NH-CH-R), 3.67 (d, J = 6.2 Hz, 1H, (NH-CH-R), 3.63 (s, 1H, O-CH3). 13C NMR (101 MHz, CDCl_3_) δ 171.76, (NH-CO), 167.97, (O-C=O), 157.61, (C-O-CH3), 135.74, 133.79, 133.59, 130.22, 129.86, 129.81, 129.37, 129.31, 129.07, 128.93, 128.73, 128.60, 128.51, 127.48, 125.57, 120.80, 110.26, 75.90, (O-(CH)CO), 55.10, (O-CH3) 39.95, (NH-(CH2)R. HR-MS (ESI): [M+H^+^] calculated for C_23_H_21_NO_4_: 376.1435, found: 376.1560.
**N.** **(4-methoxybenzylcarbamoyl)phenyl)methyl benzoate**

Prepared from 1-(isocyanomethyl)-4-methoxybenzene (0.15 g, 1 mMol), benzaldehyde (0.11 mL, 1 mMol), and benzoic acid (0.122 g, 1 mMol) according to the general procedure. Purification: column chromatography on silica gel (3:1 DCM/hexane). Isolated as a white powder (98% yield); mp 112–114 °C; 1H NMR (400 MHz, CDCl_3_) δ 8.72 (s, 1H, NH), 8.09–7.93 (m, 2H, Ar-H), 7.47 (dt, J = 15.6, 9.4 Hz, 1H, Ar-H), 7.43–7.31 (m, 2H, Ar-H), 7.26–7.16 (m, 1H, Ar-H), 7.08 (dd, J = 14.6, 9.1 Hz, 1H, Ar-H), 6.80–6.55 (m, 1H, Ar-H), 6.25 (d, J = 30.0 Hz, 1H, (O-(CH)CO), 4.69 (s, 1H, OCH3), 4.62 (s, 1H), 4.36–4.28 (m, 1H, (NH-CH-R), 3.68–3.62 (m, 1H, (NH-CH-R). 13C NMR (101 MHz, CDCl_3_) δ 164.35, (NH-CO), 161.83, (O-C=O), 159.17, 143.09, 133.71, 130.46, 130.19, 129.53, 129.26, 129.02, 128.88, 128.49, 127.81, 127.48, 123.36, 114.46, 114.33, 114.18, 114.13, 113.97, 76.08, (O-(CH)CO), 55.26, (OCH3), 41.88, (NH-CH2). HR-MS (ESI): [M+H^+^] calculated for C_23_H_21_NO_4_: 376.1402, found: 376.2408.
**O.** **ethyl 2-[2-(benzoyloxy)-2-phenylacetamido]benzoate**

Prepared from ethyl-2 isocyanobenzoate (0.18 g, 1 mMol), benzaldehyde (0.11 mL, 1 mMol), and benzoic acid (0.122 g, 1 mMol) according to the general procedure. Purification: column chromatography on silica gel (3:1 DCM/hexane). Isolated as a white powder (88% yield); mp 130–132 °C; IR (NaCl) v(cm^−1^) 3242 (NH), 1730 (CO) ester, and 1683 (CO) amide. 1H NMR (400 MHz, CDCl_3_) δ 12.02 (s, 1H, NH), 8.65 (d, J = 8.6 Hz, 1H, Ar-H), 8.27 (d, J = 7.5 Hz, 2H, Ar-H), 7.93 (dd, J = 14.6, 8.6 Hz, 1H, Ar-H), 7.59 (d, J = 7.2 Hz, 2H Ar-H), 7.53 (t, J = 7.4 Hz, 1H, Ar-H), 7.48–7.36 (m, 4H, Ar-H), 7.37–7.24 (m, 3H, Ar-H), 7.01 (t, J = 7.6 Hz, 1H, Ar-H), 6.38 (s, 1H, O-(CH)CO), 4.37 (q, J = 7.1 Hz, 1H, CH), 4.31–4.19 (m, 2H, CH2), 1.29 (t, J = 7.1 Hz, 3H, CH_3_). 13C NMR (101 MHz, CDCl_3_) δ 168.12 O-C=O benzoate), 167.48 (NH-CO), 165.09 (O-C=O ester), 140.87, 135.72, 134.55, 133.54, 131.30, 130.75, 129.24, 128.85, 128.37, 127.24, 123.05, 120.54, 115.87, 62.02, 61.44, 14.18. HR-MS (ESI): [M+H+] calculated for C_25_H_21_NO_6_: 431.1317, found: 431.1020.
**P.** **(naphthalen-3-ylcarbamoyl)(phenyl)methyl benzoate**

Prepared from 1-isocyanonaphthalene (0.15 g, 1 mMol), benzaldehyde (0.11 mL, 1 mMol), and benzoic acid (0.122 g, 1 mMol) according to the general procedure. Purification: column chromatography on silica gel (3:1 DCM/hexane). Isolated as a white powder (80% yield); mp 171–172 °C; IR (NaCl) v(cm^−1^) 3422 (NH), 1713 (CO) ester, and 1623 (CO) amide. 1H NMR (400 MHz, DMSO) δ 8.15 (d, J = 7.4 Hz, 1H, (NH), 7.99–7.90 (m, 1H, Ar-H), 7.85 (d, J = 7.3 Hz, 1H, Ar-H), 7.80 (d, J = 8.0 Hz, 1H, Ar-H), 7.77–7.67 (m, 1H, Ar-H), 7.59 (dd, J = 9.7, 5.6 Hz, 2H, Ar-H), 7.57–7.51 (m, 1H, Ar-H), 7.52–7.43 (m, 1H, Ar-H), 6.53 (s, 1H, O-(CH)CO). 13C NMR (101 MHz, DMSO) δ 168.14, (NH-CO), 165.81, (O-C=O), 136.03, 134.25, 134.19, 133.11, 130.03, 129.62, 129.44, 129.34, 129.25, 128.66, 128.63, 127.93, 126.61, 126.49, 126.00, 122.98, 122.83, 76.44, (O-(CH)CO). HR-MS (ESI): [M+H^+^] calculated for C_25_H_20_NO_3_: 382.1339, found: 382.1316.
**Q.** **(2-iodophenylcarbamoyl)phenyl)methyl benzoate**

Prepared from 1-iodo-2-isocyanobenzene (0.23 g, 1 mMol), benzaldehyde (0.11 mL, 1 mMol), and benzoic acid (0.122 g, 1 mMol) according to the general procedure. Purification: column chromatography on silica gel (3:1 DCM/hexane). Isolated as a green solid (85% yield); mp 104–106 °C; IR (NaCl) v(cm^−1^) 3249 (NH), 1730 (CO) ester, and 1679 (CO) amide. 1H NMR (400 MHz, CDCl_3_) δ 8.45 (s, 1H, NH), 8.24 (d, J = 8.3 Hz, 1H, Ar-H), 8.18 (d, J = 7.2 Hz, 2H, Ar-H), 7.82 (d, J = 8.0 Hz, 1H, Ar-H), 7.69 (d, J = 6.7 Hz, 2H, Ar-H), 7.57 (t, J = 7.6 Hz, 4H, Ar-H), 7.43 (t, J = 7.7 Hz, 3H, Ar-H), 7.27 (dd, J = 12.1, 4.8 Hz, 2H, Ar-H), 7.08–7.01 (m, 1H, Ar-H), 6.79 (td, J = 7.9, 1.5 Hz, 2H, Ar-H), 6.46 (s, 1H, O-(CH)CO). 13C NMR (101 MHz, CDCl_3_) δ 166.92, (NH-CO), 164.85, (O-C=O), 139.58, 138.85, 137.32, 135.20, 133.97, 130.48, 130.20, 129.45, 129.35, 129.14, 128.88, 128.69, 127.67, 127.47, 126.43, 121.66, 76.13, O-(CH)CO). HR-MS (ESI): [M+H^+^] calculated for C_21_H_16_INO_3_: 458.0158, found: 458.0715.
**R.** **(benzo[d]thiazol-2-ylcarbamoyl)(phenyl)methyl benzoate**

Prepared from 2-isocyanobenzo[d]thiazole (0.16 g, 1 mMol), benzaldehyde (0.11 mL, 1 mMol), and benzoic acid (0.122 g, 1 mMol) according to the general procedure. Purification: column chromatography on silica gel (3:1 DCM/hexane). Isolated as a white solid (63% yield); mp 122–124 °C; IR (NaCl) v(cm^−1^) 3286 (NH), 1710 (CO) ester, and 1679 (CO) amide. 1H NMR (400 MHz, CDCl_3_) δ 9.01 (s, 1H, NH), 8.09 (d, J = 7.4 Hz, 1H, Ar-H), 7.96 (d, J = 7.4 Hz, 1H, Ar-H), 7.79 (d, J = 7.6 Hz, 1H, Ar-H), 7.61 (t, J = 7.5 Hz, 1H, Ar-H), 7.57–7.50 (m, 1H, Ar-H), 7.45 (dd, J = 16.4, 8.4 Hz, 2H, Ar-H), 7.33 (t, J = 7.6 Hz, 1H, Ar-H), 7.25 (t, J = 9.0 Hz, 1H, Ar-H). 13C NMR (101 MHz, CDCl_3_) δ 165.18, (NH-CO), 164.31, (O-C=O), 161.31, (Ar-C=S), 133.52, 132.17, 130.96, 130.76, 129.56, 129.23, 128.95, 128.80, 128.06, 127.93, 126.72, 125.26, 123.07, 122.07, 120.68, 120.50, 119.74. HR-MS (ESI): [M+H^+^] calculated for C_22_H_17_N_2_O_3_S: 389.0861, found: 389.2442.
**S.** **3-(3-methoxyphenylcarbamoyl)-5-bromo-2-oxoindolin-3-yl 4-bromobenzoate**

Prepared from 1-isocyano-3-methoxybenzene (1.33 mL, 1 mMol), 5-bromo isatin (0.23 g, 1 mMol), and 4-bromobenzoic acid (0.21 g, 1 mMol) according to the general procedure. Purification: column chromatography on silica gel (3:1 DCM/hexane). Isolated as a yellow liquid (90% yield); IR (NaCl) v(cm^−1^) 3286 (NH), 1710 (CO) ester, and 1679 (CO) amide. 1H NMR (400 MHz, CDCl_3_) δ 9.09 (s, 1H, (NH-amide)), 8.75 (s, 1H, (NH-indole)), 7.98 (dd, J = 14.9, 7.7 Hz, 3H, (Ar-H), 7.49 (dd, J = 12.7, 7.2 Hz, 2H, (Ar-H)), 7.35 (dd, J = 16.1, 8.2 Hz, 6H, (Ar-H)), 7.16 (dt, J = 23.6, 8.0 Hz, 3H, (Ar-H)), 6.97 (dd, J = 17.5, 8.8 Hz, 2H, (Ar-H)), 6.85 (d, J = 7.8 Hz, 1H, (Ar-H)), 6.62 (dd, J = 8.2, 1.7 Hz, 1H, (Ar-H)), 5.18 (s, NH-indole), 3.68 (s, 3H, (O-CH3)), 2.08 (s, 1H). 13C NMR (101 MHz, CDCl_3_) δ 173.55, (-CO-indole), 171.03, (-CO-ester), 163.84, (CO-amide), 161.87, (C-OCH3), 160.17, 142.31, 138.36, 134.36, 133.41, 131.12, 130.14, 129.77, 129.61, 128.75, 128.46, 128.08, 124.91, 124.85, 123.44, 112.37, 111.47, 111.22, 105.66, 55.39, 31.00. HR-MS (ESI): [M+H^+^] calculated for C_23_H_18_Br_2_N_2_O_6_: 575.9202, found: 575.6714.
**T.** **3-(p-tolylcarbamoyl)-5-bromo-2-oxoindolin-3-yl 4-bromobenzoate**

Prepared from 1-isocyano-4-methylbenzene (0.13 mL, 1 mMol) 5-bromo isatin (0.23 g, 1 mMol), and 4-bromobenzoic acid (0.21 g, 1 mMol) according to the general procedure. Purification: column chromatography on silica gel (3:1 DCM/hexane). Isolated as yellow liquid (76% yield); IR (NaCl) v(cm^−1^) 3244 (NH), 1718 (CO) ester, and 1682 (CO) amide. 1H NMR (400 MHz, DMSO) δ 11.12 (d, J = 40.1 Hz, 1H, NH-amide), 10.09 (d, J = 23.3 Hz, 1H, NH-indole), 8.71 (d, J = 11.1 Hz, 1H, (Ar-H), 8.23 (s, 1H, Ar-H), 7.87 (d, J = 8.4 Hz, 1H, Ar-H), 7.77–7.61 (m, 1H, Ar-H), 7.47 (d, J = 8.3 Hz, 2H, Ar-H), 7.09 (dd, J = 17.5, 8.0 Hz, 3H, Ar-H), 6.87 (d, J = 8.3 Hz, 1H, Ar-H), 2.25 (s, 3H, CH3). 13C NMR (101 MHz, DMSO) δ 167.13 (CO-indole), 162.97 (CO-amide), 159.82 (CO-ester), 159.49, 140.46, 136.22, 133.16, 133.00, 132.17, 131.77, 130.25, 129.70, 127.37, 119.52, 118.10, 114.71, 20.92. HR-MS (ESI): [M+H^+^] calculated for C_23_H_16_Br_2_N_2_O_4_: 541.9416, found: 541.4483.
**U.** **3-(2-nitrophenylcarbamoyl)-5-bromo-2-oxoindolin-3-yl 4-bromobenzoate**

Prepared from 1-isocyano-2-nitrobenzene (0.15 g, 1 mMol), 5-bromo isatin (0.23 g, 1 mMol), and 4-bromobenzoic acid (0.21 g, 1 mMol) according to the general procedure. Purification: column chromatography on silica gel (3:1 DCM/hexane). Isolated as a yellow liquid (85% yield); IR (NaCl) v(cm^−1^) 3280 (NH), 1712 (CO) ester, and 1676 (CO) amide. 1H NMR (400 MHz, DMSO) δ 11.07 (s, 1H, NH-amide), 8.20 (d, J = 8.2 Hz, 1H NH-indole), 7.93 (d, J = 8.6 Hz, 3H, Ar-H), 7.88–7.80 (m, 3H, Ar-H), 7.73 (t, J = 7.6 Hz, 1H, Ar-H), 7.66 (d, J = 8.4 Hz, 1H, Ar-H), 7.55 (t, J = 7.7 Hz, 1H, Ar-H), 7.49–7.41 (m, 6H, Ar-H), 7.36 (t, J = 7.7 Hz, 4H, Ar-H), 7.01 (t, J = 6.8 Hz, 4H, Ar-H), 6.87 (t, J = 11.5 Hz, 1H, Ar-H), 6.58 (t, J = 7.7 Hz, 3H, Ar-H). 13C NMR (101 MHz, DMSO) δ 184.86 (-CO-indole), 172.49 (-CO-ester), 167.11 (-CO-amide), 159.84 (-C-NO2), 151.16, 146.68, 144.17, 138.74, 136.10, 135.54, 132.08, 131.71, 131.44, 130.69, 130.43, 130.40, 127.35, 126.12, 125.80, 125.09, 123.15, 119.62, 118.23, 115.85, 112.63. HR-MS (ESI): [M+H+] calculated for C_22_H_13_Br_2_N_3_O_6_: 575.9320, found: 575.6714.
**V.** **3-(4-chlorophenoxy)-N-(naphthalen-1-yl)-2-oxoindoline-3-carboxamide**

Prepared from 1-isocyano-naphthalein (0.15 mL, 1 mMol), isatin (0.147 g, 1 mMol), and 4-chlorophenol (0.13 g, 1 mMol) according to the general procedure. Purification: column chromatography on silica gel (3:1 DCM/hexane). Isolated as a yellow liquid (80% yield); IR (NaCl) v(cm^−1^) 3295 (NH), 1053 (CO), and 1679 (CO) amide. 1H NMR (400 MHz, DMSO) δ 11.06 (s, 2H, NH-amide), 10.34 (s, 1H, NH-indole), 9.71 (s, 1H, Ar-H), 8.59 (d, J = 10.5 Hz, 1H, Ar-H), 8.49 (s, 1H, Ar-H), 8.22–8.05 (m, 1H, Ar-H), 7.98 (dd, J = 25.7, 7.5 Hz, 1H, Ar-H), 7.83 (dd, J = 26.9, 7.8 Hz, 1H, Ar-H), 7.72 (dd, J = 16.1, 7.8 Hz, 1H, Ar-H), 7.58 (t, J = 7.6 Hz, 3H, Ar-H), 7.50 (d, J = 7.3 Hz, 2H, Ar-H), 7.19 (d, J = 8.8 Hz, 2H, Ar-H), 7.06 (t, J = 7.5 Hz, 2H, Ar-H), 6.91 (d, J = 7.9 Hz, 2H, Ar-H), 6.77 (d, J = 8.8 Hz, 2H, Ar-H). 13C NMR (101 MHz, DMSO) δ 184.87 (CO-indole), 160.79 (CO-amide), 159.85, 156.76, 151.15, 138.83, 134.11, 133.03, 129.60, 128.80, 126.59, 126.53, 126.13, 125.25, 125.16, 123.22, 122.76, 122.23, 119.78, 118.29, 117.37, 112.65. HR-MS (ESI): [M+H+] calculated for C_22_H_13_Br_2_N_3_O_6_: 573.93, found: 573.80.
**W.** **3-(4-chlorophenoxy)-2-oxo-N-p-tolylindoline-3-carboxamide**

Prepared from 1-isocyano-4-methylbenzene (0.13 mL, 1 mMol), isatin (0.147 g, 1 mMol), and 4-chlorophenol (0.13 g, 1 mMol) according to the general procedure. Purification: column chromatography on silica gel (3:1 DCM/hexane). Isolated as a yellow liquid (87% yield); IR (NaCl) v (cm^−1^) 3259 (NH), 1048 (CO), and 1680 (CO) amide. 1H NMR (400 MHz, DMSO) δ 11.07 (s, 1H, NH-amide), 10.11 (s, 1H, NH-indole), 9.72 (s, 2H, Ar-H), 8.71 (d, J = 11.1 Hz, 1H, Ar-H), 8.24 (s, 1H, Ar-H), 7.58 (t, J = 7.7 Hz, 1H, Ar-H), 7.55–7.41 (m, 2H, Ar-H), 7.19 (d, J = 8.7 Hz, 4H, Ar-H), 7.14–6.99 (m, 3H, Ar-H), 6.91 (d, J = 7.9 Hz, 1H, Ar-H), 6.77 (d, J = 8.7 Hz, 4H, Ar-H), 2.25 (s, 3H, CH3). 13C NMR (101 MHz, DMSO) δ 184.87 (CO-indole), 162.96 (CO-amide), 159.86, 159.81, 156.76, 151.16, 138.82, 136.22, 133.17, 133.00, 130.25, 129.69, 129.60, 125.16, 123.22, 122.77, 119.54, 118.29, 118.12, 117.37, 112.65, 20.92. HR-MS (ESI): [M+H^+^] calculated for C_22_H_17_ClN_2_O_3_: 573.0911, found: 393.2902.
**X.** **3-(4-chlorophenoxy)-N-(3,5-dimethylphenyl)-2-oxoindoline-3-carboxamide**

Prepared from 1-isocyano-3,5-dimethylbenzene (0.131 mL, 1 mMol), isatin (0.147 g, 1 mMol), and 4-chlorophenol (0.13 g, 1 mMol) according to the general procedure. Purification: column chromatography on silica gel (3:1 DCM/hexane). Isolated as a yellow liquid (87% yield); IR (NaCl) v(cm^−1^) 3283 (NH), 1050 (CO), and 1679 (CO) amide. 1H NMR (400 MHz, DMSO) δ 11.07 (s, 1H, NH-indole), 10.05 (s, 1H, NH-amide), 9.72 (s, 2H, Ar-H), 8.75 (d, J = 11.0 Hz, 1H, Ar-H), 8.23 (s, 1H, Ar-H), 7.59 (t, J = 7.7 Hz, 1H, Ar-H), 7.50 (d, J = 7.4 Hz, 1H, Ar-H), 7.19 (d, J = 8.8 Hz, 6H, Ar-H), 7.07 (t, J = 7.5 Hz, 1H, Ar-H), 6.91 (d, J = 7.8 Hz, 1H, Ar-H), 6.78 (t, J = 8.7 Hz, 5H, Ar-H), 6.71 (s, 1H, Ar-H), 2.23 (s, 5H, CH3). 13C NMR (101 MHz, DMSO) δ 184.87 (CO-indole), 159.94 (CO-amide), 159.86, 156.76, 151.16, 138.83, 138.57, 138.35, 129.60, 125.60, 125.17, 123.23, 122.77, 118.29, 117.38, 117.27, 115.63, 112.65, 21.52. HR-MS (ESI): [M+H^+^] calculated for C_23_H_19_ClN_2_O_3_: 406.1136, found: 406.4155.

## 4. Conclusions

In this work, we have shown the reactivity, efficiency, and reusability of immobilized sulfuric acid on silica gel. This solid catalyst is applicable in MCRs as a heterogenous catalyst. The Passerini reaction was assayed in the presence of SiO_2_-H_2_SO_4_ for the first time. Our findings also demonstrated that water is a good substitute for volatile organic solvents, which is in line with one of the green chemistry principles and important for future applications. The following are a few benefits of this protocol: quick reaction times, high product yields, a simple workup, mild reaction conditions, environmental compatibility, and a lack of side products.

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
