# Peer review of "Immobilized Sulfuric Acid on Silica Gel as Highly Efficient and Heterogeneous Catalyst for the One-Pot Synthesis of Novel α-Acyloxycarboxamides in Aqueous Media"

_ijms, 2022, doi:10.3390/ijms23179529_

Round 1
Reviewer 1 Report
The ms is on the use of H2SO4/silica catalyst for the aqueous synthesis of acyloxycarboxamides.
Au-s should revise:
- The abstract is too detailed re precise reaction times. The term "Passerini" reaction is used twice. The word "retrievable" should be replaced.
- p 3: lines 79 and 80 should be united.
- Scheme 1: insert temperature. In the text, RT should be specified between 18-35 C.
- Table 1: in case the yields are only 32-65%, what is the remaining part?
- Scheme 2: rt should be specified.
-Table 2 cannot appear in the present form. The caption "reaction condition" is not relevant. The list of the reaction components is repeated 5 times, that is awkward.
- Table 3 is not too informative, the time is the same (10 min). The table should be converted into text.
- p 7, Figure 2: pls eliminate individual structures. A general scheme is desirable. The yields should be given only in a range.
- Scheme 4: the arrow symbole is not OK, cross the arrow in the middle by double slanting line.
- Scheme 5: convert into a general scheme with substitution pattern.
- Scheme6 is confusing, pls give a usual scheme arrangement with general structures and substitution pattern.
-General Procedure: give also g or ml quantities for all, even the changing components.
- the compound characterization is badly missing. This is a serious shortcoming.
In summary, a major revision is requested before acceptance in IJMS.
Author Response
Reviewer: #1
The abstract is too detailed re precise reaction times. The term "Passerini" reaction is used twice. The word "retrievable" should be replaced.
Thank you so much for this comment. The abstract has been revised as you rightly observed.
p 3: lines 79 and 80 should be united.
Thank you. The two pages have been united.
- Scheme 1: insert temperature. In the text, RT should be specified between 18-35 C.
The temperature has been added to scheme, and in the text.
- Table 1: in case the yields are only 32-65%, what is the remaining part?
The remaining part, according to the NMR Characterization we did indicate some starting materials such as benzoic acid.
- Scheme 2: rt should be specified.
Corrected as observed.
Table 2 cannot appear in the present form. The caption "reaction condition" is not relevant. The list of the reaction components is repeated 5 times, that is awkward.
Thank you for the comment. As suggested, the reaction condition has been removed from the table.
- Table 3 is not too informative; the time is the same (10 min). The table should be converted into text.
Thank you for your observation. Table 3 has been converted into text based on your suggestion, and more information has been added.
- p 7, Figure 2: pls eliminate individual structures. A general scheme is desirable. The yields should be given only in a range.
Thank you. As you rightly pointed out. The individual structures have been deleted and are now represented with a general scheme.
Scheme 4: the arrow symbole is not OK, cross the arrow in the middle by double slanting line.
Thanks for that observation. Amended as suggested.
- Scheme 5: convert into a general scheme with substitution pattern.
Thanks for the comment. The scheme has been converted into a general scheme with a substitution pattern.
- Scheme 6 is confusing, pls give a usual scheme arrangement with general structures and substitution pattern.
The scheme has been converted into a general scheme with a substitution pattern.
-General Procedure: give also g or ml quantities for all, even the changing components.
Amended as rightly suggested.
- the compound characterization is badly missing. This is a serious shortcoming.
Am sorry. This was a mistake. The missing MS has been added to the supplementary documents.
In summary, a major revision is requested before acceptance in IJMS.
Thanks so much for your time and contribution.
Reviewer 2 Report
In this paper the authors describe the immobilization of sulfuric acid on silica and its utilization in the multicomponent reaction, the Passerini reaction. The authors found that performing the reaction in water could be done in a short time and after optimization they used different substrates. The work is interesting but some points should be clarified. First, is the catalytic reaction occurred under homogeneous or heterogeneous conditions? Is there any leaching? Leaching experiments should be done. The catalyst was not identified, at least IR and determination of the immobilized acid should be performed with analytical methods. More cycles should be added to the recycling experiments and the catalyst should be analyzed after reactions. After each cycle the loading of the immobilized sulfuric acid should be determined. In addition, a control experiment using only silica as catalyst should performed. What is the pH of the water used in this work? An experiment using water as medium without the catalyst should be performed.
Author Response
Reviewer: #2
The work is interesting, but some points should be clarified. First, is the catalytic reaction occurred under homogeneous or heterogeneous conditions? Is there any leaching? Leaching experiments should be done.
Thank you very much for the comment. The catalytic reaction occurs under heterogeneous conditions. We do not observe any occurrence of leaching.
The catalyst was not identified, at least IR and determination of the immobilized acid should be performed with analytical methods.
Thanks so much for that. We have now identified the catalyst via the IR and PXRD. The details are contained in the manuscript.
More cycles should be added to the recycling experiments and the catalyst should be analyzed after reactions.
Thank you very much for the comment. The recycling experiment has been further increased to five turns, as you rightly suggested. Also, the recovered catalyst after five cycles was analyzed via IR and PXRD.
In addition, a control experiment using only silica as catalyst should performed.
Thanks so much for that. The control experiment using only silica in aqueous media was carried out; the result is contained in the main manuscript.
What is the pH of the water used in this work?
The water used was distilled water and Ph IS 6.8.
An experiment using water as medium without the catalyst should be performed.
Thank you for the comment. The experiment was done initially, and the result is in table 1.
Round 2
Reviewer 1 Report
The revised ms file does not contain the corrections. Pls highlight! It is a very trivial obligation during a revision.
Where is the comment on the low yields?
As regards the Experimental details, the compound characterizations should appera in the min body of the ms.
Author Response
The revised ms file does not contain the corrections. Pls highlight! It is a very trivial obligation during a revision.
Am so sorry. I don't know what went wrong. I submitted the track changes of all your addressed comments. However, I have highlighted them in red color in the main manuscript. Only the figures are not highlighted but have been changed to reflect the general scheme and substitution pattern as you rightly suggested in the first comment.
Where is the comment on the low yields?
Thank you so much. The low yield is associated with solubility issues. It has been established that passerini reactions perform well in some solvents such as DCM and MeOH. I have also indicated this in the main manusript.
As regards the Experimental details, the compound characterizations should appear in the min body of the ms.
Thank you for the comment. The compound characterizations have been copied to the main manuscript, as you rightly suggested.
Reviewer 2 Report
The authors have addressed properly the comments of this reviewer
Author Response
The authors have properly addressed the comments of this reviewer.
Thanks so much for your valuable contribution.